# Cross sectional analysis of clinical trials search results for cancer patients using a navigator-assisted clinical trials search using five different search engines

Milica Paunic[1‡☉], Sanghyuk Rim[2☉], Olla Hilal[3], Renée Nassar[4], Zoe Driedger[5], Farwa Zaib[2], Kayla Touma[6], Mahmoud Hossami[6‡], Rhonda Abdel-Nabi[6‡], Roaa Hirmiz[4], Caroline Hamm[2,7☉¤*]

1 Department of Medicine, Temerty School of Medicine, University of Toronto, Toronto, Ontario, Canada, 2 Department of Medicine, Schulich School of Medicine & Dentistry, Western University, Windsor, Ontario, Canada, 3 Department of Medical Sciences, Western University, London, Ontario, Canada, 4 Clinical Trials Navigator Program, Windsor, Ontario, Canada, 5 Department of Psychology, University of Windsor, Windsor, Ontario, Canada, 6 Department of Biomedical Sciences, Master of Science in Translational Health Science Program, University of Windsor, Windsor, Ontario, Canada, 7 Department of Oncology, Schulich School of Medicine & Dentistry, Western University, Windsor, Ontario, Canada

☉ These authors contributed equally to this work.
‡ These authors also contributed equally to this work.
¤ Current address: Windsor Cancer Centre Foundation, 2220 Kildare Rd, Windsor, Ontario, Canada N8W 2X3
* caroline.hamm@wrh.on.ca

## Abstract

### Background

Clinical trials play a critical role in providing patients with access to novel treatments and therapies. However, limitations in clinical trial search engines impede healthcare professionals and patients from accessing the most suitable clinical trials. This study aimed to address this issue by conducting a critical analysis of several prominent clinical trial search websites, including ClinicalTrials.gov, Canadian Cancer Trials, Clinical Trials Ontario, Canadian Cancer Clinical Trials Network, and Q-CROC.

### Methods

To identify areas for improvement, three skilled clinical trials navigators independently curated clinical trial searches for 18 cancer patients over a 2-month period. After verifying patients' eligibility for enrollment in clinical trials, the navigators documented their search outcomes and identified several limitations in the current search engines.

### Results

Careful curation of clinical trials for 18 patients revealed 247 trials. However, 140 eligible trials out of 247 (57% with 95% binomial confidence interval [50%, 63%]) were

**Data availability statement:** All relevant data are within the manuscript and its Supporting Information files.

**Funding:** Canadian Cancer Clinical Trials Network C.H. No applicable grant number. https://3ctn.ca/. The funders had no role in study design, data collection and analysis, decision to publish, or preparation of the manuscript WeSpark Health Institute C.H. No applicable grant number. https://www.wesparkhealth.com/. The funders had no role in study design, data collection and analysis, decision to publish, or preparation of the manuscript Cancer Research Collaboration Fund C.H. No applicable grant number. https://wecf.ca/community-engagement/cancer-research-collaboration-fund/. The funders had no role in study design, data collection and analysis, decision to publish, or preparation of the manuscript TD Bank C.H. No applicable grant number. https://www.td.com/us/en/about-us/communities/ready-commitment/funding-opportunities/td-charitable-foundation. The funders had no role in study design, data collection and analysis, decision to publish, or preparation of the manuscript.

**Competing interests:** The authors have declared that no competing interests exist.

found only on alternative websites yet not discoverable on the initial ClinicalTrials.gov searches, even though they were listed on ClinicalTrials.gov. Our study revealed multiple deficiencies in available clinical trials search engines. Lack of reliability was repeatedly identified in all search engines.

## Discussion

This study highlights that the current clinical trial search system needs improvement to enhance patient outcomes. It needs to be highlighted that these searches were performed by trained and dedicated clinical trials navigators. The challenges facing patients and health care professionals in navigating would be much greater. The findings from this study can serve as a foundation for the development of enhanced search engines with improved functionality, which will enable healthcare professionals and patients to find and access the most suitable clinical trials with greater ease and accuracy.

## Introduction

According to the American Society of Clinical Oncology (ASCO), every person with cancer should have the opportunity to participate in a clinical trial [1]. Clinical trials offer access to cutting-edge therapies and are the gold standard in the establishment of new and effective treatments [2]. Better patient outcomes are reported in hospitals that are involved in trials [3]. As well, patient groups with lower participation in clinical trials have reported lower improvements in overall survival [3,4]. Enrolling patients at a higher rate advances treatment at a faster rate allowing for concurrent survival to increase [5].

Despite clinical trials being the gold standard for finding reliable treatments, it is estimated that only 3% of adult cancer patients participate in clinical trials [6]. Lack of trial availability is the most common barrier to trial recruitment especially in non-academic hospitals since they tend to hold fewer within-center trials compared to larger centers. Lack of patient accrual to trials leads to trial failures. One in four clinical trials fails to recruit enough patients increasing costs of clinical trials and causing ethical compromises to patients already enrolled [7].

Most patients and health care professionals (HCPs) must rely on finding trials via databases such as ClinicalTrials.gov. A Clinical Trials Navigator program was established in 2019 to help both patients and HCPs find clinical trials [8]. In the current study we report the challenges of using five clinical trials search engines used by our navigators in finding trials for patients in Canada.

### Background

ClinicalTrials.gov is an online repository of ongoing and completed trials, including clinical trials, across the world. It was founded in February 2000 after the FDA Modernization Act was passed into law in November 1997 and has since been maintained

by the National Library of Medicine at the National Institute of Health (NIH). It was initially designed to provide the largest publicly accessible bank of information on specific studies using specific interventions and investigational products [9,10]. ClinicalTrials.gov data can be accessed in many ways including a basic field search and an advanced search option that contains over 20 structured fields such as trial status, trial type, trial phase, cancer type, funder type and trial location. The site was developed to fulfill Section 113 of the Act which requires the National Institute of Health to "establish a registry of clinical trials for both federally and privately funded trials of experimental treatments for serious or life-threatening diseases or conditions" [9,11]. It is important to note that ClinicalTrials.gov does not account for all clinical trials internationally, however, it does serve as one of the largest to-date databases. It was not until September 2007 that a new user interface was developed, representing the first significant change to the website since its launch in 2000 including the implementation of the advanced search option [12]. A second update to the site was recently completed. Studies that critically analyze clinical trial search engines such as ClinicalTrials.gov are sparse in the literature despite these repositories being the main avenue for clinical trial search.

Strategies suggested to improve clinical trial accrual include implementing clinical trials navigators which has been recently explored [8]. In 2019, the national Clinical Trials Navigator pilot program (CTN), sponsored by the Canadian Cancer Clinical Trials Network (3CTN) [13], was launched to help cancer patients, especially those outside of larger academic hospitals, find suitable clinical trials [8]. During the summer of 2022, the CTN program began this analysis. The CTN program conducted searches to find eligible clinical trials for cancer patients referred to our program using the ClinicalTrials.gov database [14]. In addition to ClinicalTrials.gov, four other online databases were searched such as Canadian Cancer Trials [15], Clinical Trials Ontario [16], Canadian Cancer Clinical Trials Network [13], and Q-CROC (Quebec – Clinical Research Organization in Cancer) [17] which include trials that are specific to Canada, Ontario, Canada, and Quebec respectively. Navigators registered and categorized challenges in the search processes and results.

Functional and user-friendly clinical trials search engines are central to successful clinical trials accrual. Poor functionality would hinder patients and healthcare providers from finding the most suitable clinical trials and potentially impair patient outcomes, as well as impair rapid clinical trials accrual leading to increasing costs to the entire system. In addition, equitable access to clinical trials is impaired if the available clinical trials search engines are not user-friendly.

## Objectives

The objective of this study was to critically analyze reproducibility and reliability of five different clinical trials search engines that are currently available to patients in Canada. We hypothesized that we would find a lack of reproducibility and consistency between the different clinical trials search engines.

We set out to critically analyze current clinical trial search engines. Our research team examined current clinical trials search engines by comparing the following five different search platforms available for cancer patients in Canada: ClinicalTrials.gov, Canadian Cancer Trials, Clinical Trials Ontario, Canadian Cancer Clinical Trials Network, and Q-CROC to identify the challenges in using these systems and investigate options to develop more efficient trial searches. For proper analyses, use of patient data, standardized search terms, an expertise of websites' structure and language, the organization of study records, and repetition of searches were required.

## Methods

Patients who had subscribed to the CTN program were included in this study. Sequential patients who were registered between July 6, 2022 and September 14, 2022 were selected for this analysis. We opted not to extend the study duration beyond September due to the implementation of new clinical trial navigators at the conclusion of this timeframe. This decision was made to maintain consistency in participant interaction and data collection methodologies throughout the study. In addition, having roughly received 100 patient referrals a year, the 18 patients were considered a reasonable proportion of the yearly total.

Patients recruited to the CTN program via the Canadian Cancer Clinical Trials Network website, physician or patient referral, or advertisements in the pilot site and subscribed to our service via an online portal [8]. Patients or their representatives completed a written consent form, which led them to an on-line survey. Data collected from this patient population was analyzed. For the purpose of this study, all patient information was de-identified and no author would be able to identify patients during or after data collection.

The Clinical Trial Navigators Program (CTN) is a pilot program launched in 2019 to help Canadian cancer patients find suitable clinical trials by providing personnel or navigators to conduct clinical trial searches for patients. Three navigators at the CTN host site were hired to search for clinical trials for cancer patients. The number of clinical trial navigators was limited to three as at the time of the research project, we only had three trained navigators to perform the clinical trial searches. Each navigator underwent onboarding training to provide them with the correct tools to be able to conduct searches. Training modules included a clinical trials module, cancer terminology (including stage, biomarkers, treatment options), examination of the previous results of the clinical trials searches, the rationale and history of the clinical trials navigator program and TCPS2–2022 training. Weekly meetings and review of the clinical trials searches allowed for ongoing improvement in clinical trial search expertise. Each navigator has developed disease site expertise as well to increase the ability of each navigator to identify the maximal available clinical trials. During the onboarding process, it became evident that the navigators were discovering significant challenges in the clinical trials search engines available. The navigators in this study included one Clinical Trials Associate and two university student undergraduate students. The data was analyzed by the PI, a medical oncologist, the entire of the team of navigators (medical students, the Clinical Trials Associate, and undergraduate university students).

## Study design

We developed standardized methods to examine these challenges within the clinical trial websites over a 2-month period. We performed clinical trials searches for 18 different patients with cancer diagnoses. Three trained clinical trials navigators repeated their searches independently on all 18 patients. All three would first search ClinicalTrials.gov and next, would search an alternate website to find eligible trials for the same patients. They recorded all trials found initially on ClinicalTrials.gov and then recorded the trials identified on second websites that were not found on the initial ClinicalTrials.gov search. Then, they investigated this second group of studies and logged how many were registered on ClinicalTrials.gov even though were not initially discoverable on ClinicalTrials.gov. Search terms were defined by the individual navigators and were not pre-determined. Repeated searches were performed on the same websites to investigate the reproducibility of the searches.

Here, we perform a quantitative analysis of 18 patient search results across 5 different websites done by 3 different navigators. Patient searches that were not done by all three navigators were not included to ensure consistent data collection. All three navigators performed searches for all 18 patients independently. Each navigator was provided with the same referral form that had the pertinent patient information needed to determine trial eligibility including the type and stage of cancer, date of birth, the previous and current line of therapies with dates, and outcomes. All patient referral forms were de-identified. The only patient identifiers were their initials and date of birth. Using this information, each navigator independently searched 2–3 websites: All navigators searched ClinicalTrials.gov plus one to two alternative websites. Navigator 1 searched ClinicalTrials.gov, 3CTN, and Q-CROC. Navigator 2 searched ClinicalTrials.gov and Ontario Clinical Trials. Navigator 3 searched ClinicalTrials.gov and Canadian Cancer Trials. For every search conducted on ClinicalTrials.gov, the navigators used the 'advanced search' function to conduct searches and each kept these filters constant: "Interventional Studies" for study type and "Recruiting" for study status. Trial locations considered for eligibility were found in Canada as well as Bethesda, Maryland through the National Institute of Health Research (NIH). For each patient, each navigator provided a report of eligible trials. This was followed by a second review by the primary investigator of the CTN team, and the report is then provided to the referring physician and/or the patient.

After every patient search, each navigator filled out a shared spreadsheet for data collection. All patients that used the CTN program were included consecutively. Each navigator kept track of the search terms used for every ClinicalTrials.gov search. For all clinical trial websites searched, each put down the number of trials for the following categories: total studies identified by the search engine, total number of eligible trials, total eligible trials in Ontario and total eligible exclusively in Quebec, as well as total eligible uniquely in Bethesda through NIH. Total number of trials were defined as the number of trials populated by the search engines. For each search done on each website, each navigator would analyze each clinical trial independently and check its inclusion and exclusion criteria to determine if the patient is eligible for that trial. This determined the number of eligible trials.

For the navigator-specific alternative websites, each navigator noted the number of trials found on alternative websites that were not present on their initial search through ClinicalTrials.gov yet that still existed in the ClinicalTrials.gov database. These were identified by a second search of the ClinicalTrials.gov website using the identifiers provided by the alternative website. In addition, any trials found eligible on the alternative websites were double-checked for recruiting status since some trials listed as recruiting were no longer recruiting. As well, all searches were standardized to the same geographic region of the alternative website. Fig 1 summarizes the procedure followed during the study period for data collection by each navigator.

No Equity, Diversity, Inclusion (EDI) information was collected during this 2-month study, however, the program has implemented a collection of EDI information going forward. Unavoidable delays were experienced in PLOS ONE delaying the publication of this paper, originally submitted in 2023.

## Statistical analysis

By the end of the study, two analyses were conducted. One analyzed the variability in the number of trials found on ClinicalTrials.gov during the initial search. The second analyzed the difference between the number of eligible trials found on each website for the same patient. This was done to analyze how many eligible trials were found on alternative websites that existed yet were not discoverable in the initial ClinicalTrials.gov search.

To analyze the variability in the number of trials found on ClinicalTrials.gov during the initial search, a zero-inflated Poisson random effects model was performed with navigator and patient as random effects. To compare the proportion of trials identified on ClinicalTrials.gov versus alternative websites, the observed proportion with a 95% binomial confidence interval was reported.

In addition to quantitative data collection, qualitative data was collected. Throughout the search process, each navigator noticed ambiguities in their search results and reported challenges found in the search engines with specific examples. During the analysis, the reported challenges were categorized into the following: lack of reproducibility, lack of discoverable trials, lack of standardized and inclusive search terms; errors in the search engines and Inadequate or out of date information to complete a meaningful trials search (including lack of adequate filters such as prior therapies). These categories were determined through an iterative approach of examining the challenges recorded by the research team and designed to include all concerns raised during the research process [18].

Study size was determined by sequentially choosing all patients registered within a two-month period to the CTN program. This would represent 17% of our annual patient population. Each patient had three unique clinical trials searches performed by different navigators leading to 54 separate clinical trials searches.

In order to control for variability in the skills of the different navigators, we also compared the skill sets of the different navigators by statistically analyzing the results of the individual searches. Minor variations in search terms were identified and this could confound our results.

## Patient selection

Patients who had subscribed to the CTN program during the summer of 2022 accrual period were included in this study. Research assistants worked at the host site of the CTN program.

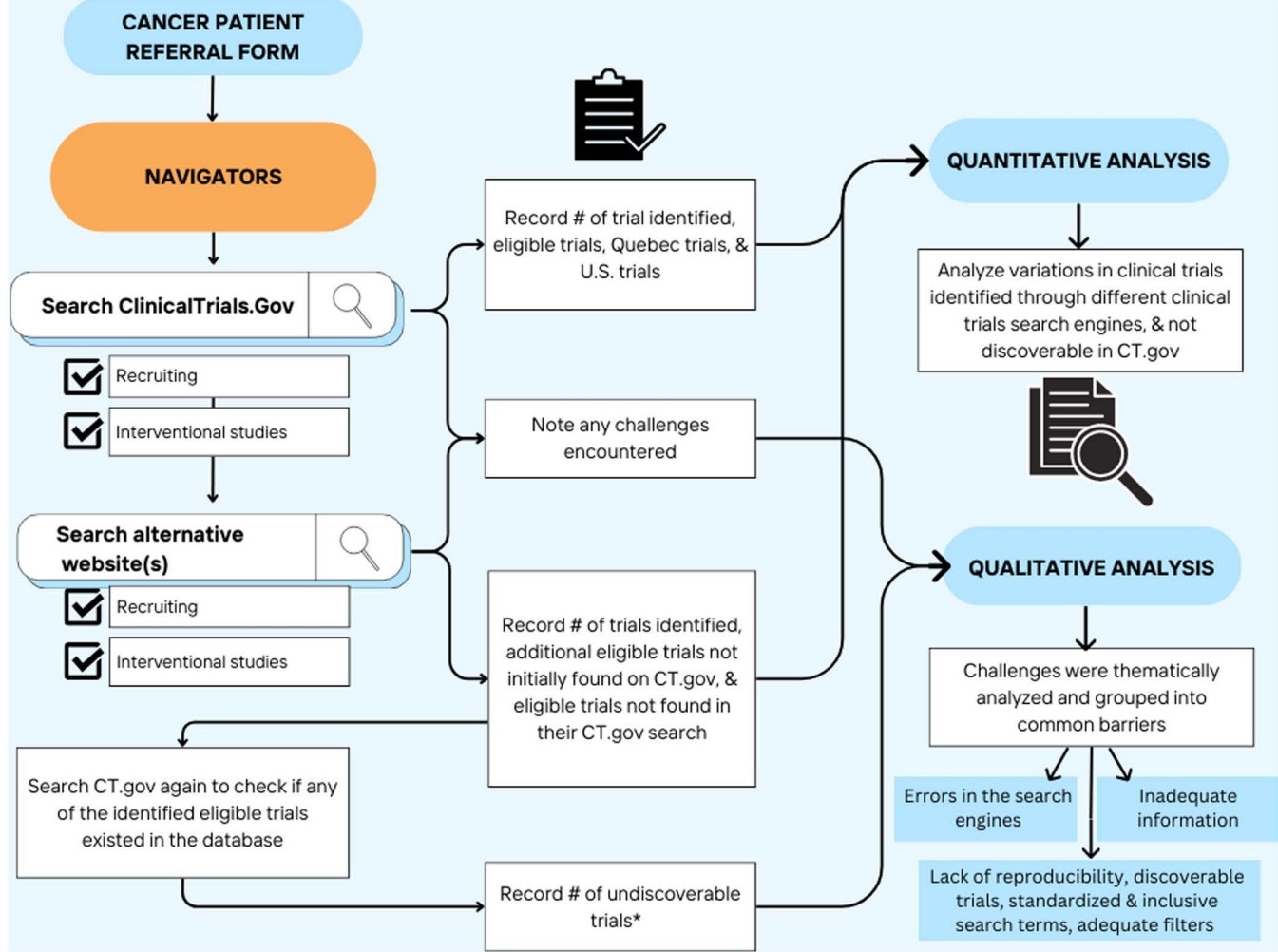

**Fig 1. Graphical representation of the methodologies.** CT.gov = ClinicalTrials.gov. *Undiscoverable trials = trials not found on the initial ClinicalTrials. gov search but found on alternative website and present on the ClinicalTrials.gov website.

This research was approved by the Research Ethics Board at Windsor Regional Hospital #22–439, Category A approval.

## Results

We examined eighteen cancer patients registered in the CTN who had had searches performed by three different navigators. The median age of these patients referred to the CTN program was 67 (31–81) years of age. 56% of the referred patients were male, 39% were female, and one patient lacked gender information. Out of the 18 patients, 56% had a stage 4 or relapsed refractory cancer with the most common cancers occurring in greater than 10% of referred patients being pancreatic cancer, breast cancer, colorectal cancer, and brain cancer (11% pancreatic cancer, 11% breast cancer, 17% colorectal, and 28% brain cancer). The median number of prior lines of therapy was 1.5 (0–12). The CTN program allowed patients to be referred more than once if they wanted another clinical trial search. For instance, patient 6 and 7 are the exact same patient just with a repeated search done over a month apart from the first. The characteristics of the study population are shown in Table 1.

**Table 1. Demographics.**

| Demographics | |
| --- | --- |
| Median Age | 67 (31-81) |
| Gender | 10 Males (56%) |
| | 7 Females (39%) |
| | 1 Unknown (6%) |
| Median # prior lines of therapy | 1.5 (0-12) |
| Cancers occurring in greater than 10% | 2 Pancreatic cancer (11%) |
| | 2 Breast cancer (11%) |
| | 3 Colorectal cancer (17%) |
| | 5 Brain cancer (28%) |
| Disease Site | |
| Hematological malignancies (acute leukemia, lymphoma, myelodysplastic syndrome, multiple myeloma) | N = 4 (22%) |
| Gastrointestinal malignancies (small bowel adeno-carcinoma, pancreatic, colorectal) | N = 6 (33%) |
| Brain tumour | N = 5 (28%) |
| Breast cancer | N = 2 (11%) |
| Melanoma | N = 1 (6%) |

In the process, we identified several challenges in using the currently available search engines for cancer clinical trials. We categorized these challenges into the following categories: lack of reproducibility, undiscoverable trials, lack of standardized and inclusive search terms, inadequate or out of date information to complete a meaningful trials search (including lack of adequate filters such as prior therapies), and search engines errors.

## Lack of reproducibility

Our study identified a lack of reproducibility of search results on ClinicalTrials.gov, Canadian Clinical Trials and Ontario Clinical Trials. Searches done by the same navigator on the same browser using the same search filters led to a different number of studies shown in two identical searches conducted within a 5-minute period. Similarly, refreshing the search within a small timeframe of a few minutes would lead to change in results, both resulting in a loss and gain in studies, despite the search filters staying consistent after refreshing. In addition, navigators 2 and 3 conducted a search for a patient where they used the same search terms just in a different sequence and still received a different number of reported studies. As well, the use of the 'Next Study' function on ClinicalTrials.gov, used to scroll through the identified list of potential trials, changed the number of eligible total trials.

Searches conducted by different navigators led to variability in not only the total number of studies but also the number of eligible studies, even when applying similar or identical search terms. Three navigators performed searches for all patients on ClinicalTrials.gov using virtually identical search terms. Out of a total of 54 ClinicalTrials.gov searches, only 5 searches revealed identical clinical trials results between navigators and four of these were when zero trials were found. For other patients, the variation in the number of trials found in total ranged from zero to seventeen between the different searches. Fig 2 below highlights that results can vary significantly due to the ClinicalTrials.gov search engine despite using similar search terms.

To control for the various skill sets of the different navigators, we assessed the efficacy of the different navigators in using ClinicalTrials.gov. The median number of trials identified for each patient by each navigator is shown in Fig 3. Navigator 1 had a median of 3.5 trials identified per patients (range: 0–10), navigator 2 had a median of 5 trials identified

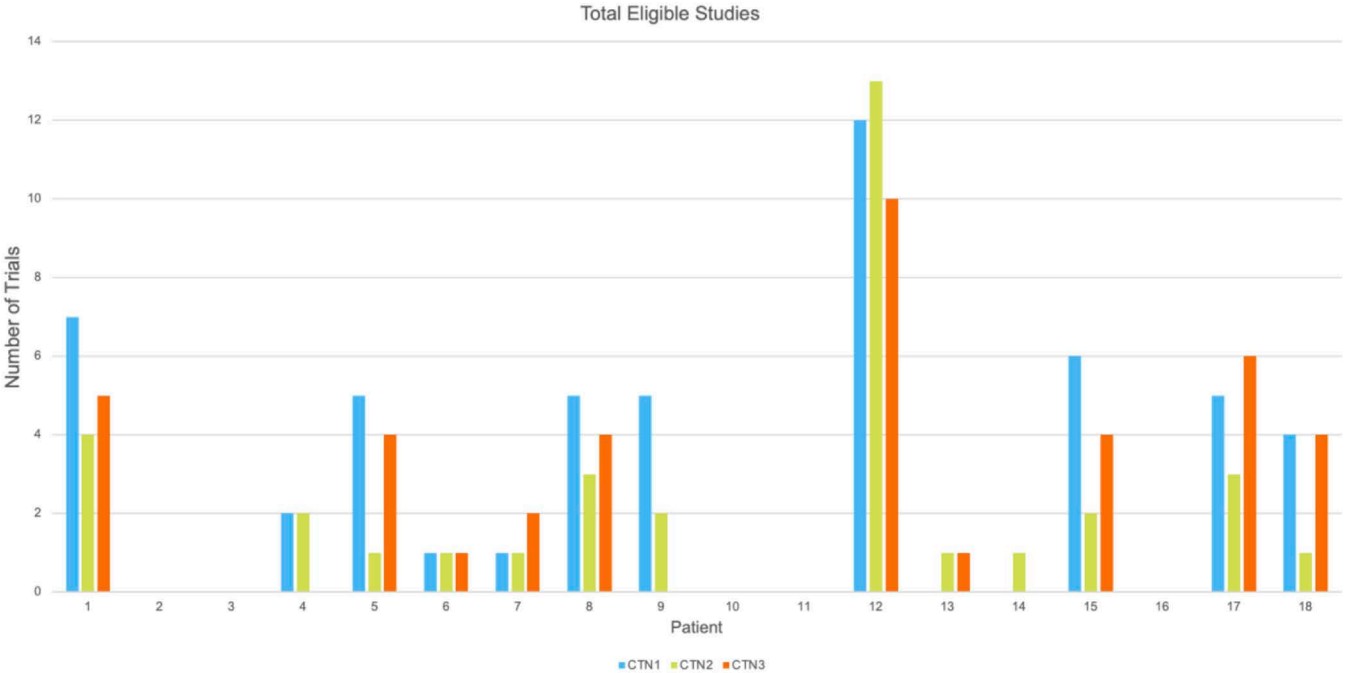

**Fig 2. Comparison of the number of eligible trials found on the initial ClinicalTrials.gov search per patient done by each navigator searching with the same patient information.** CTN1, CTN2, and CTN3 represent clinical trials navigator 1, clinical trials navigator 2, and clinical trials navigator 3 respectively.

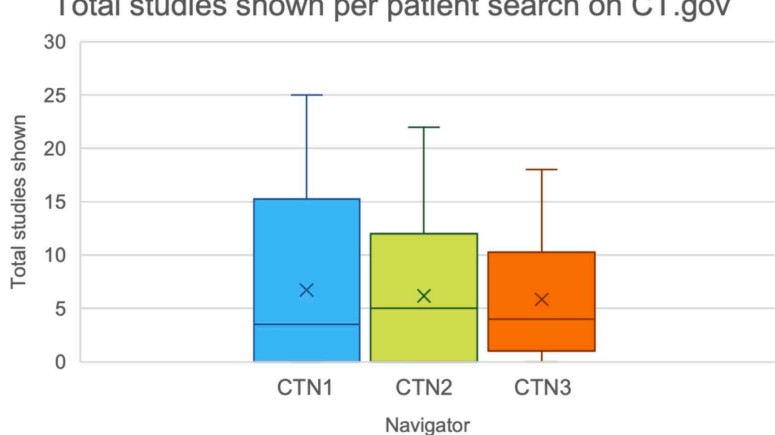

**Fig 3. The median number of trials identified on ClinicalTrials.gov by each navigator.** CTN1, CTN2, and CTN3 represent clinical trials navigator 1, clinical trials navigator 2, and clinical trials navigator 3 respectively.

(range: 0–13), and navigator 3 had a median of 4 trials identified per patient (range: 0–12). A zero-inflated Poisson generalized linear mixed model with navigator and patient as random effects was fit the data to provide a more accurate representation of the variability in the number of trials found per patient. The intraclass correlation coefficients (ICCs) from navigator is around 0% indicating that navigator-related variability is negligible. While ICC from patients is 57.9%, suggesting that differences between patients contribute significantly to the variation in the number of trials found. These results

highlight that inconsistencies in trial identification are primarily driven by patient-level factors rather than navigator-specific search behavior. This is an expected result, as each patient will have different demographics and eligibility.

**Undiscoverable trials**

In addition, we had difficulties in identifying all potential studies on the ClinicalTrials.gov website. Many navigators found eligible ClinicalTrials.gov trials for patients on alternate websites, but not on the initial ClinicalTrials.gov search. If we had not searched the alternative websites, many of the eligible trials would have been undiscovered. The results are presented in Fig 4. The first bar in Fig 4 shows that in the searches done for the 18 patients by clinical trial navigator 1, they identified 31 Canadian trials as eligible from their initial ClinicalTrials.gov search and then found 29 additional eligible trials on 3CTN, the alternative website. The second bar demonstrates that in the searches conducted by navigator 1 again, 25 Quebec trials were identified as eligible from their initial ClinicalTrials.gov search and 66 additional eligible trials were then found in QCROC search engine. The 66 trials found on this alternate website, similarly to the other results, existed in the ClinicalTrials.gov database but were not discovered during the first search on ClinicalTrials.gov. The third bar demonstrates that in the searches conducted by clinical trials navigator 2, 17 Ontario trials were identified as eligible from their initial ClinicalTrials.gov search and 7 additional eligible trials were then found in Ontario Clinical Trials. The fourth bar shows that in the searches done by clinical trials navigator 3, 34 Canadian trials were identified as eligible from their initial ClinicalTrials.gov search while 38 additional eligible trials were then found in Canadian Cancer Trials. These 38 trials were available on the ClinicalTrials.gov website but not discoverable on the first search. To summarize, 140 eligible trials out of 247 (57% with 95% binomial confidence interval [50%, 63%]) were identified by the navigators and discoverable on alternative websites yet not discoverable on the initial ClinicalTrials.gov searches. Although all 140 trials existed in the ClinicalTrials.gov database, they were not identified by the navigators when conducting their first search on ClinicalTrials.gov and would have not been reported to patients if the navigators had not searched the alternative websites.

Each CTN column reflects eligible studies found in the initial ClinicalTrials.gov search relative to those found on the alternate website. For example, in the first bar, the navigator identified 31 clinical trials in ClinicalTrials.gov and 29 trials in Clinical Trials Ontario (CTO). Yet, all 60 of these listed trials were registered on ClinicalTrials.gov but 29 were not discoverable by our trained navigators.

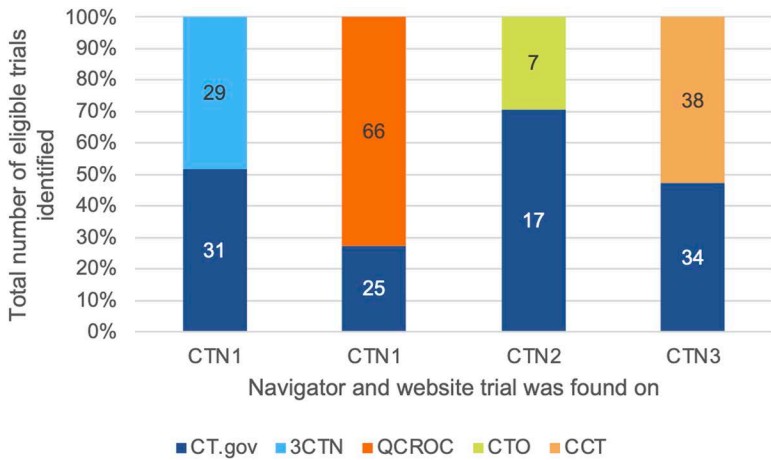

**Fig 4. The per cent eligible trials identified on ClinicalTrials.gov versus on the alternative website that each navigator searched.** CTN1, CTN2, and CTN3 represent clinical trials navigator 1, clinical trials navigator 2, and clinical trials navigator 3 respectively. CT.gov = ClinicalTrials.gov, 3CTN = Canadian Cancer Clinical Trials Network, QCROC = Quebec – Clinical Research Organization in Cancer, CTO = Clinical Trials Ontario, and CCT = Canadian Cancer Trials.

### Lack of standardized and inclusive search terms

Various synonyms can be used to search for clinical trials, but the search engine is not controlled for these synonyms. Searching for stage IV disease will produce different results than searching for metastatic cancer; while searching for a glioblastoma multiforme patient, switching only "GBM" for "Glioblastoma Multiforme" in the search query yielded a different number of studies shown on websites.

### Inadequate or out of date information to complete a meaningful trials search (including lack of adequate filters such as prior therapies)

Many trials on the alternative websites lacked a full description of exclusion and inclusion criteria. The navigator would have to perform a second search for the specific trial on ClinicalTrials.gov to determine the eligibility of the patient. The websites never provided important details such as which cohort of the study is currently open. This is especially important if the patient is only eligible for one cohort of the study. Other studies which were no longer recruiting were listed as open to accrual. These issues were only identified after the patient was referred for clinical trial, and extensive discussion with the trial's sites identified these issues.

Our study identified a complete lack of co-location filtering in ClinicalTrials.gov, 3CTN, and Canadian Cancer Trials. In other words, the clinical trials in different locations cannot be searched simultaneously, and this requires separate searches for each different location. The process of doing the same search for different locations and eliminating overlapping clinical studies increases the workload for the website users and slows down the process of finding appropriate trials for the patients. Also, setting a location as a filter generates different results depending on if the location was included in writing in the location query or by using the map feature on ClinicalTrials.gov.

### Errors in the clinical trials search engines

In addition, errors in search labels led to missing potential trials. For example, we noticed that for one search on Ontario Clinical Trials, inputting the age of 57 in the age search query resulted in zero trials populated; however, putting no age populated 4 trials that were all listed as eligible for anyone over the age of 18. In other words, the use of the search labels resulted in missed trials for the patient.

In several cases, selecting search labels during the search process generated results that did not match the categories chosen. For example, checking off "Interventional" studies yielded inclusion of non-interventional studies; searching for one type of cancer would present trials that were for a completely different cancer and thus were ineligible for the patient; information contained in the title of a study was different than the eligibility criteria misleading the navigators. At first glance, the patient was eligible but was ineligible after review of eligibility criteria.

## Discussion

Based on the five cancer clinical trials websites searched (ClinicalTrials.gov, 3CTN, Q-CROC, Ontario Clinical Trials, and Canadian Cancer Trials), our study found that the current clinical trials search system contains several challenges. We categorized the challenges into the following: lack of reproducibility, undiscoverable trials, lack of standardized and inclusive search terms, inadequate or out of date information to complete a meaningful trials search (including lack of adequate filters such as prior therapies) and errors in search engines. Due to the scoping nature of our qualitative analysis, we did not assess the relative representation of these challenges within the websites. Certain omissions, such as co-location and information on cohort status, were consistently identified. Documentation of the frequency of categorized search engine problems (lack of reproducibility, undiscoverable trials, lack of standardized search terms, inadequate or outdated information) will be pursued in a future study.

Lack of reproducibility was demonstrated repeatedly in four of five websites' clinical trial searching capabilities. Identical searches would not always lead to the same number of identified trials, excluding potential trials without explanation. We

revealed that, despite having trained navigators, the search engines were not able to provide a reproducible list of potential clinical trials for patients.

Undiscovered trials were repeatedly demonstrated on ClinicalTrials.gov. Over 50% of trials on ClinicalTrials.gov were only identified by searching other alternative websites. Undiscoverable clinical trials in the search process means lost opportunities to improve patient outcomes and improve accrual to open trials. Therefore, it is crucial for search engines to reproduce the same results regardless of the same search being conducted by a different user and using synonymous terms.

Lack of standardized and inclusive search terms was also demonstrated on ClinicalTrials.gov. When ClinicalTrials.gov was first established in 2000, its mandate was to be a repository of research endeavors [9]. Although it is now used a search engine for stakeholders searching for clinical trials, its functionality is compromised by a lack of standardized input. Researchers can enter the one equivalent search term, but the trial will only be discovered by searching all potential equivalent search terms, which is likely not feasible.

Inadequate or out-of-date information challenged the team in identifying a meaningful clinical trials list for patients. Closed trials, closed cohorts of trials, and lack of ability to search different locations led to further challenges.

We identified several errors in the search engines such as wrongly searching for non-interventional trials when interventional trials were requested or not identifying any trials for a patient with a specific age, but finding trials for anyone over the age of 18.

As well, different clinical trials navigators used similar but not always identical search terms, order of search terms and filters when performing clinical trial searches. Although search terms were sometimes not identical, they were highly similar. Again, this most likely would reflect the real-world experience of both patients and health care professionals performing clinical trials searches. Previous studies have identified that patients are required to grasp the medical terminology associated with cancer diagnosis and treatment in order to identify potential trials [19]. Furthermore, they must familiarize themselves with the general clinical trial research process and the specific requirements of each trial to assess their suitability and interest. Consequently, patients have encountered frustration in their efforts to enroll in trials [20].

The current national and international clinical trial databases are relatively new despite clinical trials being essential to the healthcare system and holistic patient survival. Despite websites such as ClinicalTrials.gov being the main repository for finding clinical trials for cancer patients in small to medium-sized centers, very little literature has reported a direct analysis of the efficiency of these databases for clinical trial searching. One article summarizes common problems and issues with ClinicalTrials.gov when using it for its original purpose as a research study repository [9]. However, our study is among the first to critically analyze the clinical trial search capability and accuracy of ClinicalTrials.gov and alternative Canadian websites for the purpose of finding trials for cancer patients.

Despite the lack of literature looking at challenges in these repositories, there are studies that agree with some of the findings of our study and present additional barriers as well [4,19–28]. Chaturvedi et al., discovered that ClinicalTrials.gov contains incorrect trial information. For instance, the name and roles of the people involved in clinical trials were missing in a notable amount of trial records, and there were also numerous variations in the names of principal investigators [29]. Similarly, Zarin et al. found that trial records were missing information and containing imprecise entries of trial information [28]. Another study points out some of the errors we found in ClinicalTrials.gov. Tse et al. found that trial records are often not updated on time by the responsible researchers. For instance, the recruitment status of clinical trials might not be correct. Moreover, clinical trials within the database have incomplete information which may further exacerbate the issues we have found [9,24]. Atkinson et al. conducted a comparative assessment of internet-based clinical trial search tools, revealing significant discrepancies in functionality and content despite their widespread availability. In addition, other studies have found that these current state-of-the-art clinical trial search engines are not easy to use [24,25]. A recurring challenge highlighted by previous studies is that the clinical trial search process presents an additional barrier, even for individuals interested in participating in clinical trials [19,26,27].

In addition to adopting a navigator-assisted search program, potential solutions to the discussed challenges involve imparting a more user-friendly interface, ensuring accurate trial information is presented, standardizing search terms, and increasing health literacy among patients.

Clinicaltrials.gov was built initially to address the need for researchers to place their trials on publicly available forums rather than its common use today as a means of locating eligible trials for patients [28]. Utami et al. illustrate that current clinical trial search engines are not easy to use, especially for individuals lacking adequate health literacy [25]. The complexity inherent in the tools necessitates users to provide extensive personal medical information and possess a nuanced understanding of specific clinical trial terminology. Consequently, the intricate nature of these tools suggests that online search tools may not sufficiently support the clinical trial recruitment process [24].This underscores the potential necessity for a different interface approach to aid this population in identifying clinical trials they may consider participating in [25]. These registries must be transformed to match the population that uses it, including healthcare providers and patients. One way is for developers to actively seek out input from persons with lived experience on how to simplify and enhance their search options [24]. In addition, developers should take the opportunity to collaborate with clinicians and their staff to identify optimal methods for incorporating clinical trial search processes into healthcare encounters. Given the difficulty in navigating the clinical trial repositories, many studies show that a different interface modality might be necessary to help healthcare providers and patients find clinical trials in which they might be willing to participate [25]. One study emphasized the importance of using a web-based interface that focuses on helping individuals with low expertise in navigating the websites find cancer-related clinical trials. They found that most of the problems were interface related and believed that their animated assistant would feel more natural to these users. They cited the Research-Based Web Design & Usability Guidelines, developed by the U.S. Department of Health and Human Services, which recommends that users be provided with simple search functions (Guideline 17.6), search templates (Guideline 17.9), and hints to improve search performance (Guideline 17.8) [30]. For example, including a definition of "interventional trials" when hovering over the dropdown menu option could significantly clear confusion for users using these search engines. Thus, a shift from a passive to a more patient-focused active model may ameliorate the problems we encountered. Utami et al. also found that misspelling by patients using these repositories was common, thus, an autocorrect feature is necessary for these registries [25].

Another area of improvement involves focusing on some of the specific challenges we encountered such as inaccurate trial information. Utami et al. found that if a user believed the details of participation were unclear on the registries, or if they became overwhelmed by medical terminology or even the quantity of text, the trial was rejected [25]. Challenges like these hinder patient accrual and subsequent patient survival. Rigorous guidelines should be implemented by registries so that investigators and sponsors make diligent efforts to submit complete, timely, accurate, and informative data about their studies to maximize the usefulness of websites [28]. Other studies as well recommend adding more search criteria to facilitate find a meaningful list of potential eligible trials [25]. For instance, filters we found were lacking including biomarker status or prior therapies, may be implemented to help refine searching. In addition, standardizing search terms would significantly improve the variability we found in the number of trials populated based on the words used in the search. Registries must standardize their language models so that inputting "GBM" versus "Glioblastoma Multiforme" does not populate different results.

The CTN program provides trained navigators with expertise in the medical terminology of clinical trial searching. In addition, a focus on improving the health literacy of patients has been recommended as a potential solution to mitigating the medically jargoned-repositories [24,25,28]. One study found that a keyword-based search used by these registries' engines was essentially unusable for users with inadequate health literacy [25]. Other findings show that these search engines demand a considerable level of health and computer literacy from users leading to severe losses in functionality [24]. In Utami et al.'s study, users with low health literacy found significantly fewer trials compared to those with adequate health literacy [25]. Many patients chose the wrong cancer type in the dropdown menus of certain search engines because they looked similar (i.e., clicking carcinoma of unknown primary instead of appendix carcinoma).

The small sample size is a limitation of this study. However, the challenges identified in this sample were found to be reproducible even in this small dataset. In addition, this study's search engine analyses were conducted in 2022. Since then, these platforms may have undergone updates or improvements that could impact search accuracy, accessibility, or user experience. As a result, some of the challenges identified in this study may have been mitigated, while others may persist.

Many stakeholders rely on the search engines we examined to help patients find clinical trials. Up to 80% of patients in North America are located outside of academic centers [9,31] and rely on clinical trials search engines to identify potential trials. Patients, caregivers, health care professionals, clinical trials associates, pharmaceutical employees all have a stake in a reliable search engine. As well, pharmaceutical companies, cooperative clinical trials groups and researchers rely on these search engines to identify patients for their clinical trials. They lead to faster accrual and faster discovery of new treatments for patients.

Our Clinical Trials Navigator program helps to fill a void in the field that helps patients, caregivers, and health care professionals in navigating the difficult world of performing clinical trials searches. Other studies also suggest that a clinical-trial matching service would help improve participation in clinical trials [22,32–34] . Our navigators are trained in oncology and in clinical trials searches. We demonstrated that, although there was minor variation between the navigators, all displayed similar skills in searching for clinical trials. The extra burden placed on patients, caregivers, and health care professionals in performing clinical trials searches for patients is amplified when we consider that many do not have the skill sets developed by our navigators.

We are in the process of developing novel methods to overcome these challenges in performing clinical trials searches for cancer patients. We have developed Master Lists for each large disease site that are maintained and kept up to date. We review five clinical trial search engines routinely to ensure a comprehensive list of clinical trials is created. These are used for clinical trials searches and found to be more reliable than repeating searches on clinical trials sites. In addition, the program recently implemented disease-specific navigators to increase their ability to perform meaningful and accurate searches for our clients. We are performing pilot studies to test our new methods of clinical trials searches, and these will be presented in further publications. The SQUIRE-EDU guidelines were used in preparation of this manuscript.

## Conclusion

Proper functionality of clinical trials search engines promises equitable access to clinical trials and clinical trials accrual. We reported a lack of reproducibility, undiscoverable trials, lack of standardized and inclusive search terms, inadequate or out-of-date information to complete a meaningful trials search (including lack of adequate filters such as prior therapies) and search engine failures that compromise the functionality of these search engines.

Through the Clinical Trials Navigator Program, we have developed and are testing a new process for the identification of a reproducible and meaningful clinical trials list for patients with cancer in Canada. Early results are promising and after further testing, will be reported.

## Supporting information

**S1 File. STROBE-checklist-v6-combined-PlosMedicine.**
(DOCX)

**S2 File. Raw Data.**
(XLSX)

## Acknowledgments

We would like to thank Lee McGrath for her contribution in providing administrative support, managing finances, and facilitating marketing efforts associated with this research project. Her expertise was invaluable to the success of the study. We

would also like to thank Dr. Rong Luo from the University of Windsor for their invaluable statistical expertise in analyzing the data. We acknowledge that preliminary findings were published at the WeSpark Conference held in Windsor, Ontario in November 2022, UWill Discover Conference held in Windsor, Ontario in March 2023, and the ASCO Annual Meeting in Chicago in June 2023 [35–37].

## Author contributions

**Conceptualization:** Milica Paunic, Farwa Zaib, Kayla Touma, Mahmoud Hossami, Rhonda Abdel-Nabi, Roaa Hirmiz, Caroline Hamm.

**Data curation:** Milica Paunic, Sanghyuk Rim, Olla Hilal, Renée Nassar.

**Formal analysis:** Milica Paunic.

**Funding acquisition:** Sanghyuk Rim, Caroline Hamm.

**Investigation:** Milica Paunic, Sanghyuk Rim, Olla Hilal, Renée Nassar, Caroline Hamm.

**Methodology:** Milica Paunic, Olla Hilal, Renée Nassar, Zoe Driedger, Caroline Hamm.

**Project administration:** Caroline Hamm.

**Resources:** Milica Paunic, Caroline Hamm.

**Supervision:** Caroline Hamm.

**Validation:** Milica Paunic.

**Visualization:** Milica Paunic, Sanghyuk Rim.

**Writing – original draft:** Milica Paunic, Sanghyuk Rim.

**Writing – review & editing:** Milica Paunic, Sanghyuk Rim, Caroline Hamm.

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
