## [Decision Letter · Decision Letter 0]

PONE-D-23-23984Cross-sectional analysis of clinical trials search results for cancer patients using a navigator-assisted clinical trials search using five different search enginesPLOS ONE

Dear Dr. Hamm,

Thank you for submitting your manuscript to PLOS ONE. After careful consideration, we feel that it has merit but does not fully meet PLOS ONE’s publication criteria as it currently stands. Therefore, we invite you to submit a revised version of the manuscript that addresses the points raised during the review process.

I would like to sincerely apologise for the delay you have incurred with your submission. It has been exceptionally difficult to secure reviewers to evaluate your study. We have now received three completed reviews; the comments are available below. The reviewers have raised significant scientific concerns about the study that need to be addressed in a revision.

Please revise the manuscript to address all the reviewer's comments in a point-by-point response in order to ensure it is meeting the journal's publication criteria. Please note that the revised manuscript will need to undergo further review, we thus cannot at this point anticipate the outcome of the evaluation process.

We look forward to receiving your revised manuscript.

Kind regards,

Miquel Vall-llosera Camps

Staff Editor

PLOS ONE

Journal Requirements:

2. For studies involving third-party data, we encourage authors to share any data specific to their analyses that they can legally distribute. PLOS recognizes, however, that authors may be using third-party data they do not have the rights to share. When third-party data cannot be publicly shared, authors must provide all information necessary for interested researchers to apply to gain access to the data. (https://journals.plos.org/plosone/s/data-availability#loc-acceptable-data-access-restrictions) 

Reviewers' comments:

Reviewer's Responses to Questions

**Comments to the Author**

1. Is the manuscript technically sound, and do the data support the conclusions?

Reviewer #1: Partly

Reviewer #2: Partly

Reviewer #3: Yes

2. Has the statistical analysis been performed appropriately and rigorously? 

Reviewer #1: Yes

Reviewer #2: Yes

Reviewer #3: Yes

3. Have the authors made all data underlying the findings in their manuscript fully available?

Reviewer #1: No

Reviewer #2: No

Reviewer #3: Yes

4. Is the manuscript presented in an intelligible fashion and written in standard English?

Reviewer #1: Yes

Reviewer #2: No

Reviewer #3: Yes

5. Review Comments to the Author

Reviewer #1: The criteria for not considering more clinical trial searchers are not very clear, which could provide more information to analyze the possible deficiencies that each one presents when searching for clinical trials. Similarly, the criteria for not considering a longer study time.

There is no flowchart or graphic diagram detailing the procedure or standardized methods used to analyze the results of the search for clinical trials for cancer patients.

A percentage representation of the problems encountered in the categorized search engines (lack of reproducibility, lack of searchable trials, lack of standardized and inclusive search terms, errors in the search engines, and inadequate or outdated information to complete a meaningful trial search) and a representation of the relationship between these problems and the search engines are not shown. I recommend incorporating more representations of statistical analysis.

I think you should incorporate other current studies so that you can incorporate them in the discussion of results, as well as other recommendations (based on the new incorporation) to improve the functionality of the search engines for clinical trials.

I consider that 23 bibliographic sources are too few. Perform a literature search and add results, experiences, and current trends to the search for clinical trials for cancer patients. More than 65% of the references are more than 5 years old.

Reviewer #2: This paper presents a study that addresses the limitations in clinical trial search engines by conducting a critical analysis of several prominent websites, including ClinicalTrials.gov, Canadian Cancer Trials, Clinical Trials Ontario, Canadian Cancer Clinical Trials Network, and Q-CROC. The study underscores the need for improvement in the current clinical trial search system to enhance patient outcomes. The authors discuss challenges, and their findings can potentially serve as a foundation for developing enhanced search engines with improved functionality. This advancement would enable healthcare professionals and patients to find and access the most suitable clinical trials with greater ease and accuracy. However, upon careful consideration, it appears that the content and scope of this paper may not align with the expected caliber or scholarly depth typically associated with the targeted journal for publication.

Reviewer #3: The authors have investigated the challenges in finding appropriate clinical trials for Canadian patient suffering from cancer using ClinicalTrials.gov as well as 4 other websites (Canadian Cancer Trials, Clinical Trials Ontario, Canadian Cancer Clinical Trials Network, and Q-CROC) by conducting searches for 18 patients by three trained clinical trial navigators. They identified important challenges such as lack of reproducibility, out-of-date information, undiscoverable trials, lack of standardized and inclusive search terms, and search engine failures.

Overall, this is a well-designed study addressing the important issue of reliability of these clinical trial search engines in identifying all eligible trials for patients diagnosed with cancer. Also, the manuscript is easy to read and follow. Even though this is a great effort, some questions and concerns need to be addressed:

1. The authors have assigned different alternative websites to different trial navigators. Given the relatively large variability in the number of trials found in ClinicalTrials.gov by different navigators, it seems beneficial to repeat the search of alternative website by other navigators to check the variability of findings in these websites as well. What is the reason that the search in alternative websites were not done by all navigators? Do you expect it to change the findings? If not, it needs to be discussed in the manuscript.

2. As mentioned before, it is easy to follow and read the manuscript, however it seems redundant in some parts. For example, the time interval of the study has been mentioned with details on few occasions. It can only be mentioned in the Method section or Study Design section. Removing redundant sentences in the manuscript can improve the quality of the manuscript and keep the readers engaged.

3. What are some of the changes that can potentially improve the challenges identified in this study. It worth to discuss some of these solutions in the Discussion section.

4. One of factors affecting the search outcome is the information listed in the referral forms. What information can be added to the referral forms to help with the challenges you have identified, if any? Is the level of variability in number of trials found by the navigators for each patient related to the extent of the info provided in the referral forms?

5. In line 112, the comma after “Functional” should be changed to “and”.

6. In line 293: “Out of 54 ClinicalTrials.gov searches, only 5 searches revealed identical clinical trials results between navigators and four of these were when zero trials were found”, 54 should be changed to 18.

6. PLOS authors have the option to publish the peer review history of their article (what does this mean? ). If published, this will include your full peer review and any attached files.

**Do you want your identity to be public for this peer review?** For information about this choice, including consent withdrawal, please see our Privacy Policy .

Reviewer #1: **Yes: ** Cristhian Ronceros Morales

Reviewer #2: No

Reviewer #3: **Yes: ** Mohammad Sadegh Mashayekhi

---

## [Author Response · Author response to Decision Letter 1]

25 Apr 2024

As requested, we are submitting a revised manuscript, a rebuttal letter with replies to the reviewer’s comments, and a revised manuscript with track changes.

---

## [Decision Letter · Decision Letter 1]

PONE-D-23-23984R1Cross-sectional analysis of clinical trials search results for cancer patients using a navigator-assisted clinical trials search using five different search engines

PLOS One

Dear Dr. Hamm,

Thank you for submitting your manuscript to PLOS One. After careful consideration, we feel that it has merit but does not fully meet PLOS One’s publication criteria as it currently stands. Therefore, we invite you to submit a revised version of the manuscript that addresses the points raised during the review process.

If applicable, we recommend that you deposit your laboratory protocols in protocols.io to enhance the reproducibility of your results. Protocols.io assigns your protocol its own identifier (DOI) so that it can be cited independently in the future. For instructions see: https://journals.plos.org/plosone/s/submission-guidelines#loc-laboratory-protocols . Additionally, PLOS One offers an option for publishing peer-reviewed Lab Protocol articles, which describe protocols hosted on protocols.io. Read more information on sharing protocols at https://plos.org/protocols?utm_medium=editorial-email&utm_source=authorletters&utm_campaign=protocols .

We look forward to receiving your revised manuscript.

Kind regards,

Vaishnav Khade

Peer Review Operations Specialist

PLOS One

On behalf of,

Cristhian Ronceros, Ph.D.

Guest Editor

PLOS One

Additional Editor Comments (if provided): NA

Reviewers' comments:

Reviewer's Responses to Questions

**Comments to the Author**

1. If the authors have adequately addressed your comments raised in a previous round of review and you feel that this manuscript is now acceptable for publication, you may indicate that here to bypass the “Comments to the Author” section, enter your conflict of interest statement in the “Confidential to Editor” section, and submit your "Accept" recommendation.

Reviewer #4: (No Response)

Reviewer #5: All comments have been addressed

2. Is the manuscript technically sound, and do the data support the conclusions?

Reviewer #4: Partly

Reviewer #5: Yes

3. Has the statistical analysis been performed appropriately and rigorously? 

Reviewer #4: No

Reviewer #5: Yes

4. Have the authors made all data underlying the findings in their manuscript fully available?

Reviewer #4: Yes

Reviewer #5: Yes

5. Is the manuscript presented in an intelligible fashion and written in standard English?

PLOS One does not copyedit accepted manuscripts, so the language in submitted articles must be clear, correct, and unambiguous. Any typographical or grammatical errors should be corrected at revision, so please note any specific errors here.

Reviewer #4: Yes

Reviewer #5: Yes

6. Review Comments to the Author

Reviewer #4: I found this paper an interesting read overall, and it is of interest also to those who work on systematic reviews and meta-analysis. The design is sound. statistical analysis and reporting do not fully convince me.

The authors report an analysis on difference between navigators in number of trials found per patient on clinicaltrials.gov, which they found non significant. There seems to be a typo there as they say it's not significant and then they write P<0.05, but in general I think p-values should be reported fully unless they are <0.001. Yet, in this case the problem I have with that analysis is that the small sample size, together with the apparent heteroskedasticity, weakens the results a lot. A quasipoisson or a zero-inflated poisson generalized linear model would probably work better than Kruskal Wallis and gain a bit of power, at least. In any case, it feels bizarre to read in the same paper that results were inconsistent between navigators while also reporting that they do not differ significantly. Perhaps the toolset is inadequate to communicate the point. The thing of the greatest interest in such an analysis is not whether there are differences between couples of raters, but a general measure of variability or agreement, which would make the use of the analysis of variance toolset very valuable here. By looking at the data structure, it seem that the very first data to report would be intraclass correlations for both "patient" and "navigator" as obtained from a random effect model. So, ideally, a poisson random effect model does the trick in the cleanest possible way, although a linear model would probably work too as a reasonable approximation. You cannot easily obtain p-values for random effects, but you usually don't need them as the ICCs paint a good picture already.

In general, I see this reporting of p-values even in instances where they don't really say anything relevant as the hypothesis itself is meaningless. How is it relevant to test the hypothesis that the number of trials found on other sites in less than 50% than what found on clinicaltrials.gov? It's just an arbitrary threshold, we could test if they were less than 75%, or 35%. In this case it is more sensible to just report the proprtion with the binomial confidence interval. Plus, after spending so much energy to prove that the number of trials you find on clinicaltrials.gov is not consistent across searches, treating it as a fixed number as opposed to a random variable seems a contradiction: why should we consider it a gold standard, when it has been made clear that is so severely inconsistent?

The strange thing about this paper is that it is all about repeatability of results, and yet there is zero measures of repetability or agreement, while there's plenty statistical tools made precisely for that purpose. It's perplexing and I think it should be re-worked.

As a side note, I would also add that in the particular field of research, i.e. literature searches, agreement should in theory pretty much be perfect, and even a simple description of results without any fancy statistics shows already that perfect it is not. I believe this is relevant already.

Reviewer #5: The revised manuscript titled "Cross-sectional analysis of clinical trials search results for cancer patients using a navigator-assisted clinical trials search using five different search engines" presents a thoughtful and detailed analysis of clinical trial search tools. This study addresses an important challenge: helping patients and healthcare providers find suitable clinical trials, which can often be life-changing. ClinicalTrials.gov, as one of the most widely used platforms, is highlighted as a crucial resource, but its limitations, along with those of other search tools, are well-documented in this work.

The manuscript is clear and well-organized, flowing naturally from the introduction to the conclusion. The authors have made noticeable improvements, particularly in detailing the methodology. They provide a deeper explanation of how patients were selected, how navigators were trained, and how the searches were conducted. These updates make the study more transparent and trustworthy. The added figure also help to visualize the comparisons and results more effectively.

One of the most valuable aspects of this study is its practical implications. By identifying issues such as inconsistent terminology, outdated information, and technical errors in current search tools, the authors propose clear, actionable solutions. These include expanding filters to include prior therapies, and improving user interfaces to make search engines more user-friendly. The idea of navigator-assisted searches is especially compelling, as it demonstrates a meaningful way to help patients gain better access to trials.

The authors have responded well to the reviewers’ initial comments, especially regarding inconsistencies in search results and differences between navigators.

However, I have a few additional suggestions:

1. the discussion should address why the analysis focuses on 2022 data when the manuscript was submitted in 2024. Readers might wonder about this gap, so providing a brief explanation would be helpful.

2. the repeated mention of specific dates ("July 6, 2022, and September 14, 2022") throughout the manuscript should be streamlined to avoid redundancy, as suggested by reviewer #3.

3. since the study was conducted in 2022, the authors should acknowledge that search engines might have improved since then, which is an important limitation and should be mentioned in the conclusion.

In summary, this is a well-executed and highly relevant study that provides important insights into improving clinical trial search systems. With just a few small adjustments, it will be ready for publication. The findings have the potential to benefit not only patients but also healthcare providers and policymakers working to improve access to innovative treatments.

I am delighted to serve as a reviewer for this manuscript, as it significantly contributes to my clinical practice.

7. PLOS authors have the option to publish the peer review history of their article (what does this mean? ). If published, this will include your full peer review and any attached files.

**Do you want your identity to be public for this peer review?** For information about this choice, including consent withdrawal, please see our Privacy Policy .

Reviewer #4: No

Reviewer #5: **Yes: ** Katarzyna Pogoda

---

## [Author Response · Author response to Decision Letter 2]

25 Feb 2025

We have included a response to the reviewers, a tracked change manuscript and a clean manuscript. Thank you for your time.

---

## [Decision Letter · Decision Letter 2]

PONE-D-23-23984R2Cross-sectional analysis of clinical trials search results for cancer patients using a navigator-assisted clinical trials search using five different search enginesPLOS ONE

Dear Dr. Hamm,

Thank you for submitting your manuscript to PLOS ONE. This manuscript has been recently assigned to me and after careful consideration of reviewers comments, I believe that your manuscript has merit for publication. I understand how lengthy the peer-review process has gotten. So, I moved this manuscript forward as soon as possible. Just a few points should be considered before final decision and production of the manuscript. Therefore, I give the manuscript a "minor revision", so that you are able to send the final revised version of manuscript for final decision. we invite you to submit a final revised version of the manuscript that addresses below points raised during the review process. 1- Please double check the formatting of the references to be aligned with the PLOS one referencing guideline.2- Please cite the conferences you have published a part of your work in.3- Please check the Manuscript Organization to be fully aligned with the authors guideline (about naming of headings and supplementary materials). Please submit your revised manuscript by May 30 2025 11:59PM or anytime sooner. If you will need more time than this to complete your revisions, please reply to this message or contact the journal office at plosone@plos.org . Please include the following items when submitting your revised manuscript:

We look forward to receiving your revised manuscript.

Kind regards,

Vahid Mansouri, M.D.

Academic Editor

PLOS ONE

Journal Requirements:

Reviewers' comments:

Reviewer's Responses to Questions

**Comments to the Author**

1. If the authors have adequately addressed your comments raised in a previous round of review and you feel that this manuscript is now acceptable for publication, you may indicate that here to bypass the “Comments to the Author” section, enter your conflict of interest statement in the “Confidential to Editor” section, and submit your "Accept" recommendation.

Reviewer #3: All comments have been addressed

Reviewer #4: All comments have been addressed

2. Is the manuscript technically sound, and do the data support the conclusions?

Reviewer #3: Yes

Reviewer #4: Yes

3. Has the statistical analysis been performed appropriately and rigorously? 

Reviewer #3: Yes

Reviewer #4: Yes

4. Have the authors made all data underlying the findings in their manuscript fully available?

Reviewer #3: Yes

Reviewer #4: Yes

5. Is the manuscript presented in an intelligible fashion and written in standard English?

Reviewer #3: Yes

Reviewer #4: Yes

6. Review Comments to the Author

Reviewer #3: The authors replied back to all my comments from the previous review and they were satisfying. Also, the statistical analysis is significantly improved after addressing the comments by Reviewer #4 and Reviewer #5. I think this manuscript is ready to be published.

Reviewer #4: All of my comments have been addressed. I think the paper is suitable for publication in its current state.

7. PLOS authors have the option to publish the peer review history of their article (what does this mean? ). If published, this will include your full peer review and any attached files.

**Do you want your identity to be public for this peer review?** For information about this choice, including consent withdrawal, please see our Privacy Policy .

Reviewer #3: No

Reviewer #4: **Yes: ** Alberto Ferrari

---

## [Author Response · Author response to Decision Letter 3]

22 May 2025

PONE-D-23-23984R2

Cross sectional analysis of clinical trials search results for cancer patients using a navigator-assisted clinical trials search using five different search engines.

PLOS ONE

To Vahid Mansouri, M.D.

Academic Editor

PLOS ONE

Thank you so much for your support on our paper:

1- We have double checked and the formatting of the references are aligned with the PLOS one referencing guideline.

2- Please cite the conferences you have published a part of your work in. This has been included in the ‘Acknowledgment’ section.

This manuscript has been partially presented in 2022 WE-SPARK Health Research Conference, 2023 UWill Discover Student Research Conference, and in abstract form for the American Society of Clinical Oncology, 2023.

3- The Manuscript Organization is fully aligned with the authors guideline (about naming of headings and supplementary materials).

Please find enclosed:

1. A rebuttal letter that responds to each point raised by the academic editor and reviewer(s). You should upload this letter as a separate file labeled 'Response to Reviewers'. - submitted

2. A marked-up copy of your manuscript that highlights changes made to the original version. You should upload this as a separate file labeled 'Revised Manuscript with Track Changes'. submitted

3. An unmarked version of your revised paper without tracked changes. You should upload this as a separate file labeled 'Manuscript'. Submitted

Thank you for your time.

Caroline Hamm

---

## [Editor Report · Decision Letter 3]

Cross sectional analysis of clinical trials search results for cancer patients using a navigator-assisted clinical trials search using five different search engines.

PONE-D-23-23984R3

Dear Dr. Hamm,

We’re pleased to inform you that your manuscript has been judged scientifically suitable for publication and will be formally accepted for publication once it meets all outstanding technical requirements.

Kind regards,

Vahid Mansouri, M.D.

Academic Editor

PLOS ONE
---

## [Editor Report · Acceptance letter]

PONE-D-23-23984R3

PLOS ONE

Dear Dr. Hamm,

I'm pleased to inform you that your manuscript has been deemed suitable for publication in PLOS ONE. Congratulations! Your manuscript is now being handed over to our production team.

Kind regards,

on behalf of

Dr. Vahid Mansouri

Academic Editor

PLOS ONE